# Efficacy of Contrast-Enhanced Endoscopic Ultrasonography for the Diagnosis of Pancreatic Tumors

**DOI:** 10.3390/diagnostics12061311

**Published:** 2022-05-25

**Authors:** Kensuke Yokoyama, Atsushi Kanno, Tetsurou Miwata, Hiroki Nagai, Eriko Ikeda, Kozue Ando, Yuki Kawasaki, Kiichi Tamada, Alan Kawarai Lefor, Hironori Yamamoto

**Affiliations:** 1Department of Medicine, Division of Gastroenterology, Jichi Medical University, Shimotsuke 329-0498, Tochigi, Japan; r0760ky@jichi.ac.jp (K.Y.); tetsurou_miwata@yahoo.co.jp (T.M.); m05069hn@jichi.ac.jp (H.N.); 1403ie@jichi.ac.jp (E.I.); kozue_ando@jichi.ac.jp (K.A.); kawasakiyuki1219@gmail.com (Y.K.); tamadaki@jichi.ac.jp (K.T.); yamamoto@jichi.ac.jp (H.Y.); 2Department of Surgery, Jichi Medical University, Shimotsuke 329-0498, Tochigi, Japan; alefor@jichi.ac.jp

**Keywords:** endoscopic ultrasound, contrast-enhanced endoscopic ultrasound

## Abstract

Endoscopic ultrasound can be useful for obtaining detailed diagnostic images for pancreatic disease. Contrast-enhanced harmonic endoscopic ultrasound has allowed to demonstrate not only microvasculature but also real perfusion imaging using second-generation contrast agents. Furthermore, endoscopic ultrasound fine-needle aspiration cytology and histology have become more ubiquitous; however, the risk of dissemination caused by paracentesis has yet to be resolved, and the application of less invasive contrast-enhanced endoscopic ultrasound for the differential diagnosis of pancreatic tumors has been anticipated. Contrast-enhanced harmonic endoscopic ultrasound can contribute to the differential diagnosis of pancreatic tumors.

## 1. Introduction

Ultrasonography (US) can be useful for patients with hepatic, biliary, and pancreatic diseases. This modality allows for noninvasive diagnostic imaging, with the Doppler mode being useful for assessing tumor blood flow. However, although blood flow in large blood vessels can be evaluated, evaluating blood flow in capillaries and obtaining perfusion images remain challenging. Studies have shown that contrast-enhanced ultrasonography using an ultrasound contrast agent enables the acquisition of perfusion images of parenchymatous organs [1,2,3] and is expected to contribute to the improvement of diagnostic accuracy. Despite the several reports on the use of contrast-enhanced ultrasonography for the diagnosis of pancreatic tumors [4,5,6,7,8], this approach has not gained popularity considering that the ultrasound contrast agent is not covered by Japanese public health insurance.

Endoscopic ultrasonography (EUS) has enabled detailed sonography of the pancreas, which can be diagnosed via ultrasonic observation from within the digestive tract [9,10]. Previously, EUS was performed by mechanical radial scanning; thus, blood flow data could not be obtained using the Doppler mode. However, with the advent of electronic radial EUS and advancements in ultrasound contrast agents, observation using the Doppler mode and contrast-enhanced EUS has become possible. We herein review the usefulness of EUS using contrast media for pancreatic diseases.

## 2. Contrast-Enhanced EUS

### 2.1. Contrast Agents

The contrast agent is composed of gas-filled microbubbles of 2–5 μm and the lipids or phospholipids that cover them. CE-EUS had first been by Kato in 1995 as a method of injecting CO_2_ bubbles from a celiac artery or superior mesenteric artery to evaluate the characteristic of pancreatic mass [11]. The development of the intravenous ultrasonic contrast agent Levovist^®^ (Bayer Schering Parma, Berlin, Germany), which consists of microbubbles with a diameter of about 3 μm, made possible the visualization of small blood vessels through contrast-enhanced harmonic imaging, and the qualitative diagnostic ability improved in the transabdominal US [12]. Levovist^®^ is a first-generation ultrasound contrast agent with which air microbubbles are coated with galactose and palmitic acid. This contrast agent delineates nonlinear signals generated by the collapse of air bubbles by applying ultrasonic waves with high sound pressure [2,13].

While this agent allows for Kupffer imaging and liver tumor identification, only intermittent contrasting of sound waves can be obtained, and only images of blood vessels can be depicted, such as by frame-by-frame playback. Furthermore, given that perfusion images of the pancreatic parenchyma cannot be obtained, this contrast agent may be unsuitable for obtaining contrast-enhanced ultrasound images of the pancreas. On the other hand, several second-generation ultrasound contrast agents, such as SonoVue (Bracco SpA, Milan, Italy), Sonazoid (Daiichi-Sankyo, Tokyo, Japan; GE Healthcare, Milwaukee, WI, USA), and Definity (Lantheus Medical Imaging, Billerica, MA, USA), have been developed. Sonazoid^®^ is a second-generation ultrasound contrast agent containing perflubutane of approximately 3 μm in size covered by a phospholipid film [14,15]. Similar to Levovist^®^, Sonazoid^®^ identifies liver tumors by utilizing its ability to be taken up by Kupffer cells. By delineating nonlinear signals obtained by applying ultrasonic waves with low sound pressure to resonate with the contrast agent, it is possible to obtain contrast enhancement of peripheral blood vessels and perfusion images of parenchymatous organs, which could not be obtained with Levovist^®^. Therefore, future applications of Sonazoid^®^ in contrast-enhanced ultrasonography of abdominal parenchymatous organs, such as the pancreas, has been highly anticipated. In recent years, reports have suggested the usefulness of contrast-enhanced ultrasonography from the body surface in the differential diagnosis of pancreatic diseases [4,5,6,7,8,9,10]. Notably, Kitano et al. reported on the effectiveness of contrast-enhanced ultrasonography in clearly depicting pancreatic tumors and classifying the contrast enhancement pattern for the differential diagnosis of pancreatic tumors [16]. Faccioli et al. reported that performing contrast-enhanced ultrasonography more clearly delineates the margin of pancreatic tumors and that it was useful for determining surgical indications [5]. Moreover, other reports have shown that contrast-enhanced ultrasonography can determine the viability of the pancreas prior to transplantation. However, even now, several years after Sonazoid^®^ has been made commercially available, this contrast agent is still not covered by Japanese public health insurance except in the diagnosis of liver tumors. Moreover, contrast-enhanced ultrasonography of the pancreas must be performed in clinical trials (Table 1).

### 2.2. Contrast-Enhanced Doppler Endoscopic Ultrasound

EUS has been digitized in recent years, which has enabled the delineation of Doppler images. Ultrasound contrast agents have intensified the signals of Doppler images and allowed for more clearer images of blood flow. Doppler signals have blooming artifacts, which can hinder observations; however, the eFLOW mode of the Aloka α10 (Aloka) and the H-FLOW mode of the ME2 (Olympus) can control for blooming, enabling clear delineation of blood flow images and suggesting their suitability for contrast-enhanced Doppler EUS. Some studies have utilized contrast-enhanced Doppler EUS methods for pancreatic tumor diagnosis. Accordingly, Dietrich et al., who performed contrast-enhanced EUS using the Doppler method on 93 patients with pancreatic tumors, were able to delineate hypovascular pancreatic cancer with excellent diagnosability [17]. Moreover, Hocke et al. reported a case of autoimmune pancreatitis (AIP) diagnosed using contrast-enhanced EUS with the Doppler method [18]. This enhancement in the Doppler signal by the ultrasound contrast agent is critical for determining the presence or absence of blood flow.

### 2.3. Contrast-Enhanced Harmonic Endoscopic Ultrasound

Irradiation of the ultrasound contrast agent within blood vessels using low sound pressure ultrasonic waves causes the air bubble diameter to change in accordance with the ultrasound wave cycle, generating enhanced harmonics. Contrast-enhanced harmonic endoscopic ultrasound (CEH-EUS) selectively visualizes second harmonics generated from the ultrasound contrast agent, thereby enabling perfusion images of capillaries and parenchyma. CEH-EUS allows for not only clear blood vessel imaging but also delineation of the time–intensity curve (TIC) and graphing of the changes in brightness over time through contrast.

### 2.4. CEH-EUS for Pancreatic Diseases

EUS eliminates the impact of gastrointestinal gas by performing ultrasonography from within the digestive tract. Moreover, small tumors that are difficult to identify on computed tomography (CT) scan and magnetic resonance imaging (MRI) can be delineated on EUS. This has made EUS an indispensable modality in the imaging diagnosis of pancreatic diseases. Furthermore, using second-generation ultrasound contrast agents together with the phase inversion harmonic method allows the observation of blood flow in real time, which has enabled the application of CEH-EUS in EUS. Though it is difficult to perform CEH-EUS in all patients for whom pancreatic lesions were not found using EUS, CEH-EUS is useful to diagnose pancreatic disease. Napoleon et al. performed CEH-EUS and EUS-fine needle aspiration (FNA) on 36 patients with pancreatic tumors and compared the two techniques [19]. Although CEH-EUS had inferior specificity, it had better sensitivity compared to FNA. They also showed that CEH-EUS was useful for the differential diagnosis of pancreatic tumors. Moreover, Kitano et al., who conducted CEH-EUS for pancreatic tumors, reported that it might be useful for differential diagnosis [16].

However, the aforementioned studies only evaluated the presence or absence of blood flow and patterns of contrasting but did not quantitatively evaluate the intensity of contrast enhancement. Notably, Kersting et al. who quantitatively analyzed contrast-enhanced ultrasound waves from the body surface, reported the usefulness of the same in the differential diagnosis of pancreatic cancer and mass-forming pancreatitis [10]. Moreover, Imazu et al. utilized CEH-EUS for distinguishing pancreatic cancer and AIP using the TIC [20]. Although several reports had conducted quantitative analysis, consensus regarding the methods have yet to be established [8,20]. Another study also reported of a method through which the ratio of brightness at the start of contrast enhancement divided by peak brightness is calculated and compared [21]. As such, further studies are needed to quantitatively analyze contrast-enhanced EUS.

Recently, EUS-FNA has been performed to diagnose pancreatic tumors histologically. It is difficult to differentiate between viable and necrotic tissue using only CT scan or MRI images to obtain the correct histological diagnosis of pancreatic tumors. CEH-EUS will help differentiate between viable and necrotic tissue correctly because CEH-EUS can provide sequential blood flow images in pancreatic tumors. The endosonographer will be aided by CEH-EUS images to acquire viable tissue for histological diagnosis using EUS-FNA.

## 3. Pancreatic Tumor

### 3.1. Solid Pancreatic Tumors

A normal healthy pancreas has even contrast enhancement, with the pancreatic duct delineated as an avascular tubular structure and the blood vessels delineated as renewed tubular structures. Therefore, it is easy to identify and distinguish the shape of the pancreatic duct and blood vessels using the normal fundamental B mode.

In the fundamental B mode, most solid pancreatic lesions are delineated as hypoechoic masses, making it difficult to differentiate solid pancreatic lesions. However, performing contrast enhancement makes differentiation easier. Typical pancreatic cancers present with hypo-enhancement; mass-forming pancreatitis, including AIP, often presents with iso-enhancement; and endocrine tumors often present with hyper-enhancement [22]. A report of an actual study of 277 patients with pancreatic tumors showed that the sensitivity and specificity of pancreatic cancer with hypo-enhancement were 95%, and 89%, respectively [22] (Figure 1). Furthermore, a meta-analysis of 12 studies including contrast-enhanced Doppler methods found that the sensitivity, and specificity of contrast-enhanced EUS for pancreatic cancer were 94%, and 89%, respectively, which indicated high diagnosability for pancreatic cancer [23]. Moreover, a study comparing contrast-enhanced CT with contrast-enhanced EUS revealed that contrast-enhanced EUS promoted significantly superior diagnosability for pancreatic tumors <2 cm in size [22]. In addition, EUS-FNA has been used in the histopathological diagnosis of pancreatic cancer; however, false-negative results with EUS-FNA alone pose a problem. A study of EUS-FNA combined with contrast-enhanced EUS found that EUS-FNA alone had a 92% sensitivity for detecting pancreatic cancer, whereas the addition of contrast-enhanced EUS increased the sensitivity to 100%, suggesting that contrast-enhanced EUS helps to detect false-negative cases [22].

With regard to the diagnosis of localized progression of pancreatic cancer, studies have shown that EUS alone had a diagnosability of 69%, whereas the addition of contrast enhancement increased the rate of diagnosis to 92%. More importantly, EUS alone showed a diagnosability for portal vein invasion of 83%, whereas contrast-enhanced EUS had a diagnosability of 100%, indicating the importance of contrast enhancement [24].

Reports have shown that diagnosing lymph node metastasis using the fundamental B mode is difficult; however, when using contrast enhancement, even benign cases displayed contrast enhancement, whereas cases with lymph node metastasis demonstrated uneven contrast enhancement, which can be useful for differentiation [25].

Pancreatic neuroendocrine neoplasms (pNENs) present as a well-demarcated mass on EUS. On contrast-enhanced EUS, pNENs show a diffusely uniform hypervascular pattern (Figure 2). Reports have shown that pNENs present with a hypervascular pattern on CE-EUS, with a sensitivity and specificity of 78.9% and 98.0%, respectively [26]. Furthermore, a heterogeneous enhancement pattern of pNEN has been reported to suggest malignancy [27]. However, pNEN is weakly stained by the contrast medium given the presence of severe fibrosis, making diagnosis difficult.

Solid-pseudopapillary neoplasms (SPNs) are a rare type of tumor that occurs mostly in young women. SPNs often have a solid and cystic portion with hemorrhagic necrosis. SPNs present as a hypovascular tumor compared to the surrounding pancreatic lesion on CE-EUS, with the inside of the mass being enhanced like an alveolus nest [28] (Figure 3). Kataoka reported that CEH-EUS was useful for differentiating SPNs from pNENs [29].

### 3.2. Autoimmune Pancreatitis

Studies have shown that on CEH-EUS, AIP demonstrated stronger contrast enhancement compared to tumors such as pancreatic cancer and somewhat weaker contrast enhancement compared to a healthy pancreas. Accordingly, Imazu et al. reported that CEH-EUS can be useful for distinguishing AIP and pancreatic cancer [20]. Given that ultrasound contrast agents emphasize the Doppler effect, color Doppler using contrast medium has also been reported [30]. Contrast-enhanced CT findings of AIP showed that a capsule-like rim with band-shaped low-density area forms around the pancreatic border in some instances; however, on CEH-EUS, a capsule-like rim is often delineated (Figure 4). The capsule-like rim is thought to reflect fibrosis around the pancreas, the frequency of which varies depending on the report, which can help with diagnosis.

### 3.3. Pancreatic Cystic Tumors

EUS plays an important role in the diagnosis of intraductal papillary mucinous neoplasms (IPMNs), including the diagnosis of mural nodules which is important when determining the indication for surgical resection of IPMN. However, differentiating mural nodules from mucin globs within the dilated duct(s) of IPMN remains difficult. In such instances, contrast-enhanced EUS makes it easier to differentiate mural nodules and mucin globs. One study suggested the usefulness of contrast-enhanced EUS for IPMN with internal nodules [31]. Contrast-enhanced EUS stains the septum and internal nodule areas, making it possible to distinguish them from mucous masses. Yamashita et al. also reported that CE-EUS has a sensitivity and specificity of 100% and 80% for detecting mural nodules, respectively [32]. Another report found that contrast-enhanced CT, EUS, and contrast-enhanced EUS had a diagnostic accuracy of 92%, 72%, and 98%, for mural nodules, respectively, indicating that contrast-enhanced EUS has significantly superior accuracy rates [29]. Furthermore, studies have shown that using a TIC for nodules in IPMN can promote higher intensity and reduction rates in malignant cases than in benign cases, which can be useful for differentiating benign from malignant cases [33] (Figure 5).

Serous neoplasms (SNs) are cystic tumors with a collection of micro- and macro-cysts. The area when microcysts collect in SNs appear as a hypoechoic mass on EUS. On CE-EUS, SNs present with hyper/iso-enhancement in the early phase and iso-enhancement in the late phase (Figure 6) [34].

Mucinous cystic neoplasms (MCNs) are multilocular cystic tumors with a thick fibrous cap. On EUS, the capsule is depicted as a low echo region. On CE-EUS, however, MCNs present as hyper/iso-enhancement lesions. One study reported that both SCN and MCN presented with iso/hyper-enhancement during the early and delay parenchymatous perfusion phase [34].

## 4. Summary

EUS takes on an important role in the diagnosis and treatment of pancreatic illness given its excellent resolution and enables the detection of small masses a few millimeters in size. In recent years, EUS-FNA has gained popularity, improving the histopathological diagnosability of pancreatic illness [35,36,37,38]. However, certain concerns with EUS-FNA have remained unresolved, such as the possibility of dissemination at the time of malignant tumor and pancreatic cystic neoplasm paracentesis, as well as the risk of injuring blood vessels that lie on the paracentesis route. Therefore, a noninvasive diagnostic imaging modality with a high diagnosability similarly to EUS-FNA is urgently needed.

The brightness of ultrasonography differs from that of CT. Moreover, given that the conditions vary between patients and investigations, no strict comparison can be made.

With recent technological advancements in CT, perfusion CT, through which changes in blood flow can be evaluated over time, has emerged as a useful modality. Various methodologies for perfusion CT have been established, including the maximum gradient method, compartment model method, and deconvolution method, which have been actively reported in various fields. The application of perfusion CT for pancreatic illness has also been recognized [39]. However, exposure and adverse reactions to iodine-containing contrast agents have been identified as issues for perfusion CT. Nonetheless, CEH-EUS allows for real time imaging and has superior time resolution compared to perfusion CT given that can analyze approximately 1000 images accumulated per test. Although CEH-EUS has concerns regarding the brightness level of the ultrasound waves, further developments is expected to address this issue, allowing physicians to making use of its superior properties and safety, which transcend the aforementioned shortcomings.

CEH-EUS has several disadvantages. The images from CEH-EUS tend to be affected by the technique used by each endosonographer. EUS is more invasive compared to CT scan or MRI because EUS is an endoscopic procedure. Contrast agents should be used with caution in patients with egg allergies, because one contrast agent, Sonazoid^®^ contains egg phosphatidylserine [15].

Based on available studies on CEH-EUS for pancreatic illness, as well as our experiences with the same, we believe that contrast-enhanced EUS can make use of the advantages of EUS and ultrasound contrast agents for the diagnosis of pancreatic tumors.

## Figures and Tables

**Figure 1 diagnostics-12-01311-f001:**
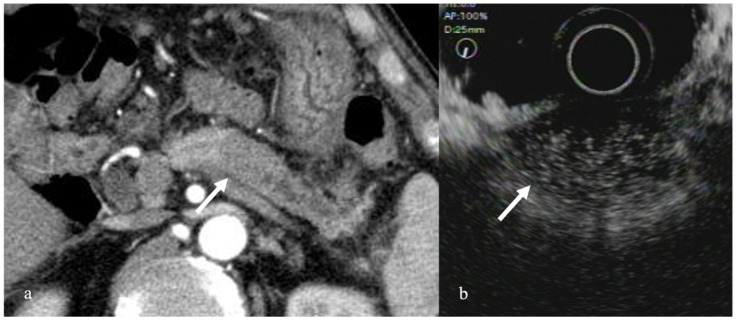
Pancreatic ductal cell carcinoma. (**a**) Contrast-enhanced computed tomography (CT) revealed a mass with poor contrast enhancement in the pancreatic body (arrow). (**b**) CEH-EUS revealed a pancreatic mass delineated as a mass with poor contrast enhancement in the contrast mode (arrow).

**Figure 2 diagnostics-12-01311-f002:**
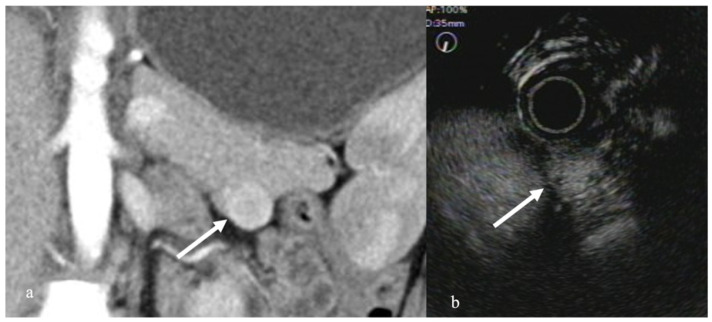
Pancreatic neuroendocrine neoplasm. (**a**) Contrast-enhanced CT revealed a hyper enhancement mass in the pancreatic tail (arrow). (**b**) CEH-EUS revealed a pancreatic mass delineated as a mass with hyper-enhancement in the contrast mode (arrow).

**Figure 3 diagnostics-12-01311-f003:**
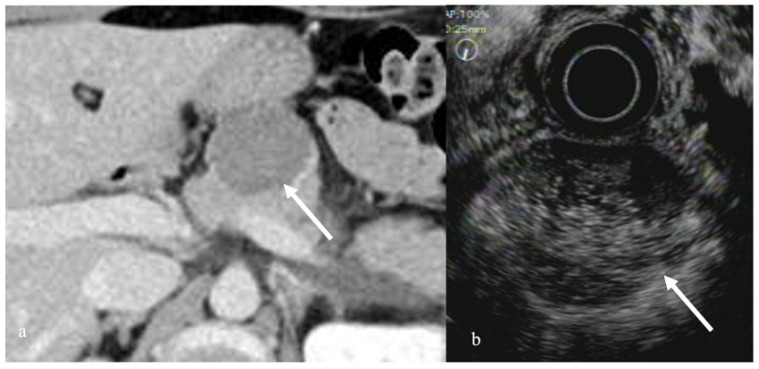
Solid-pseudopapillary neoplasm. (**a**) Contrast-enhanced CT revealed a poor contrast-enhanced tumor in the pancreatic body (arrow). (**b**) CEH-EUS revealed a pancreatic mass with poor contrast enhancement in the contrast mode (arrow).

**Figure 4 diagnostics-12-01311-f004:**
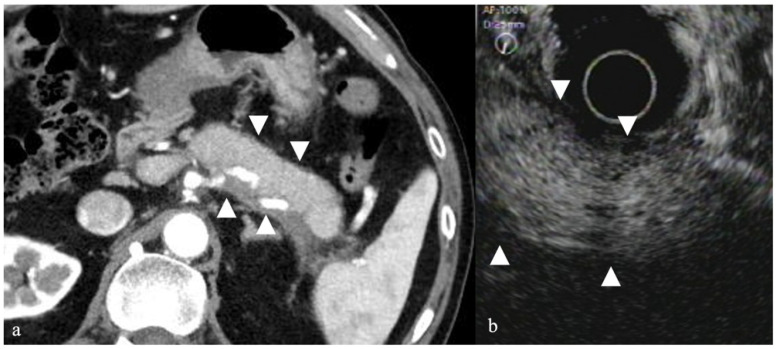
Autoimmune pancreatitis. (**a**) Abdominal CT showed a swollen pancreatic body and tail (arrowhead). (**b**) CEH-EUS revealed a strong contrast enhancement from the pancreatic center and poor contrast enhancement on the marginal region (arrowhead).

**Figure 5 diagnostics-12-01311-f005:**
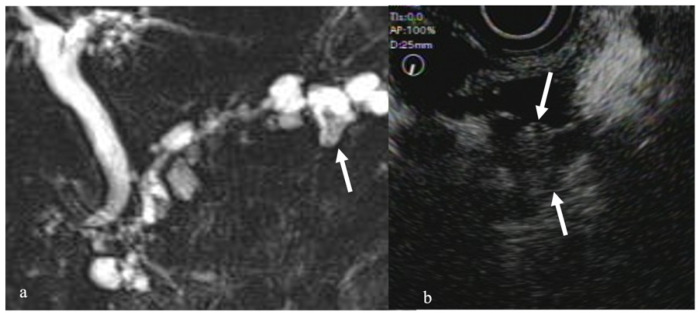
Intraductal papillary mucinous neoplasms (IPMN). (**a**) Magnetic resonance cholangiopancreatography (MRCP) revealed a dilated branch with mural nodule (arrow). (**b**) CEH-EUS revealed an enhanced nodule in the dilated branch (arrow).

**Figure 6 diagnostics-12-01311-f006:**
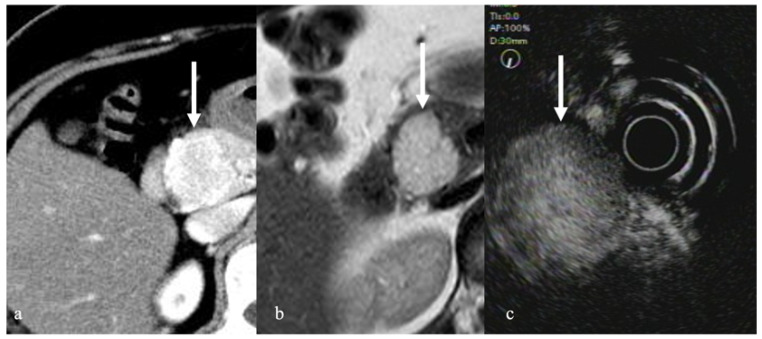
Serous neoplasm. (**a**) Contrast-enhanced CT revealed a hyper enhancement mass in the pancreatic head (arrow). (**b**) MRI with T2 weighted image showed a high intensity tumor (arrow). (**c**) CEH-EUS revealed a tumor with strong contrast enhancement in the pancreatic head (arrow).

**Table 1 diagnostics-12-01311-t001:** List of contrast agents for ultrasonography (quoted and modified from Reference [9]).

Contrast Agent	Composition
First generation	
Albunex	5% sonicated serum albumin with stabilized microbubbles
Echovist (SHU 454)	Standardized microbubbles with galactose shell
Levosist (SHU 508)	Stabilized, standardized microbubbles with galactose, 0.1% palmitic acid shell
Myomap	Albumin shell
Qantison	Albumin shell
Sonavist	Cyanoacrylate shell
Second generation	
Definity/luminity	C_3_F_8_ with lipid stabilizer shell
Sonazoid	C_4_F_10_ with lipid stabilizer shell
Imagent-imavist	C_6_F_14_ with lipid stabilizer shell
Optison	C_3_F_8_ with denatured human albumin shell
Bisphere/cardiosphere	Polylactide-coglycolide shell with albumin overcoat
Sono Vue	SF_6_ gas with lipid stabilizer shell
AI700/imagify	C_4_F_10_ gas core stabilized with polymer shell

## Data Availability

The datasets generated during the study will be available from the corresponding author on reasonable request after termination osf data collection.

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
