# Peer review of "Efficacy of Contrast-Enhanced Endoscopic Ultrasonography for the Diagnosis of Pancreatic Tumors"

_diagnostics, 2022, doi:10.3390/diagnostics12061311_

Round 1

Reviewer 1 Report

This review described the role of contrast-enhanced endoscopic ultrasonography in the diagnosis of pancreatic tumors.

The authors should add the limitation of this tool. This procedure may have been affected by the technicians. Although the authors mentioned this procedure is less invasive, I think this procedure is relatively invasive, compared to MRI and CT, although allergy may be a risk in contrast-enhanced CT and MRI.

The authors should suggest a strategy for the diagnosis of pancreatic disease including other devices. Do they recommend contrast-enhanced endoscopic ultrasonography in all patients? Fundamentally, how are the pancreatic masses (or disease) found?

Author Response

We have shown a strategy and method to find pancreatic masses and performed CEH-EUS for the diagnosis of pancreatic diseases as described on page 3, lines 107-111. It is challenging to perform CEH-EUS in all patients, not finding pancreatic lesions using EUS. As suggested by the reviewer, we added a sentence on page 3, lines 112-113 to further explain this.

To diagnose pancreatic tumors histologically, EUS-FNA has recently been used. CEH-EUS helps differentiate between viable and necrotic tissue to obtain a sufficient sample for histological tissue diagnosis. As suggested by the reviewer, we added a description of CEH-EUS in the paragraph on CEH-EUS for pancreatic diseases on pages 3-4, lines 129-134.

Reviewer 2 Report

Authors reported the review of contrast-enhanced endoscopic ultrasonography (CE-EUS) for pancreatic tumors. Authors summarized the history of contrast agents and the difference between doppler and harmonic CE-EUS. Furthermore, authors clarified the efficacy of CHE-EUS for the differential diagnosis of pancreatic tumors. I have no further comment.

Author Response

Thank you for your kind comments.

Round 2

Reviewer 1 Report

The authors have added the appropriate sentences and now I am satisfied with the acceptance.